# Features of Bread Made from Different Amaranth Flour Fractions Partially Substituting Wheat Flour

**Ionica Coţovanu *** and **Silvia Mironeasa ***

Faculty of Food Engineering, Stefan cel Mare University of Suceava, 720229 Suceava, Romania
* Correspondence: ionica.cotovanu@usm.ro (I.C.); silviam@fia.usv.ro (S.M.);
  Tel.: +40-740-816-370 (I.C.); +40-741-985-648 (S.M.)

**Featured Application:** **Particle size offers valuable information about the quality of amaranth flour, depending on milling fractions, and can be used to potentially predict the quality characteristics of new baked goods developed in the future.**

**Abstract:** Amaranth flour (AF) is recognized as high-quality raw material regarding nutrients and bioactive compounds, essential in supplying human health benefits, compared with white flour (WF). In this study, the effects of factors, different particles sizes (large, medium, and small), and levels of AF (5, 10, 15, and 20%) substituting WF on the responses, empirical and dynamic dough rheological properties, and some quality parameters of bread were successfully modeled using predictive models. Finally, the optimization of a formulation to maximize the AF level whilst maintaining bread quality for each type of particle size (PS) was performed based on the response surface methodology models generated. The rheological properties of the composite flour formulated were evaluated using Mixolab, alveograph, rheofermentometer, and dynamic rheometer. In addition, bread quality parameters, loaf volume, instrumental texture features, and firmness were evaluated. The anticipation of the optimal value for each response in terms of dough rheological properties during mixing, protein weakening, starch gelatinization and retrogradation, biaxial extension, fermentation, viscoelastic moduli, and creep and recovery compliance depending on PS. The optimal addition level was determined by a multi-objective optimization approach. The optimal addition level was 9.41% for large, 9.39% for medium, and 7.89% for small PS. The results can help manufacturers to develop bread products with the desired particle size with optimal technological and physical features.

**Keywords:** amaranth flour; bread characteristics; dough rheology; particle size; wheat flour

## 1. Introduction

Bread is one of the most consumed foodstuffs that can satisfy daily requirements and its fortification would provide an opportunity to upgrade the nutritional level of the human diet [1]. The tendency to include pseudocereals to improve the nutritional value of bakery products has become more popular [2]. The addition of amaranth flour could improve the nutritional quality of bread and bakery products because amaranth is a pseudocereal very rich in protein and carbohydrates that fulfills the needs of people with celiac disease, individuals allergic to wheat, and vegetarians [3]. The protein content of amaranth seeds is characterized by a well-balanced amino acid composition. Furthermore, amaranth seeds are rich in lipids with high levels of unsaturated fatty acids, dietary fiber, minerals (zinc, copper, and manganese), vitamins (thiamine, niacin, riboflavin, and folate), and bioactive components (squalene, tocopherols, and phenolic compounds) [4–6] with important beneficial effects on human health [7]. Moreover, scientific investigations have demonstrated that amaranth has beneficial effects on health, such as decreasing hypocholesterolemic activity, improving the immune system, decreasing blood glucose levels, acting on liver functions, and decreasing hypertension, as well as having antitumor

effects, anti-anemic effects, antioxidant activity, antiallergic actions, and beneficial effects on celiac disease [8]. The partial replacement of gluten with gluten-free flours represents a major technological challenge because gluten is the principal factor of structure-building protein, which is reflected in the quality attributes of many baked products. Furthermore, the gluten network is necessary for describing the rheological properties of dough, such as mixing ability, resistance to extension, elasticity, extensibility, and gas-holding capacity [9].

The rheological properties of dough can be evaluated by using empirical and fundamental tests. Fundamental dough rheological properties, such as oscillation, stress relaxation, and creep–recovery measurements have been previously studied to evaluate the mechanical behavior of wheat doughs [9–11], which influences the quality attributes of the end-product. In general, supplementation with non-wheat flour dilutes the gluten content in wheat flour which adversely affects the rheological properties of dough [12]. Some studies on white flour/pseudocereal composite flour highlighted that the bread properties (such as specific volume, crumb texture, and density) were positively related to dough rheological properties, and the bread's specific volume decreased as the doses of pseudocereal flour was increased [3,13,14]. Additionally, several studies demonstrated that incorporating pseudocereal flour significantly improved the nutritive values of wheat-based bakery products [15,16]. Compared with the continuous network and unique viscoelasticity of the wheat dough protein network, the protein matrix of pseudocereal flour was less desirable for bread making [17]. Including pulse flours into bread dough dilutes the gluten protein and affects both gluten development and starch–protein complexes, which are important to the dough rheology and the quality of the bread [18,19].

Mathematical modeling and statistical optimization in food processing manufacture represent a necessity in order to achieve a sustainable processing industry [20]. The response surface methodology (RSM) technique permits the study of the effects of the factors involved in processing and their interactions, using a reduced number of experimental runs, without being time-consuming.

This study aimed to optimize the formulation of amaranth flour addition levels for each particle size to enhance the dough's rheological and bread properties. To our knowledge, no other studies have examined amaranth flour's addition to wheat flour from a complex rheological properties point of view. The technological importance of each particle size could be highlighted by the chemical properties of each AF fraction, the rheological properties of the dough, and the technological properties of the bread obtained from the amaranth–wheat flour blend.

## 2. Materials and Methods

### 2.1. Materials

Wheat flour (flour yield 65%) from S.C. Mopan S.R.L., Suceava, România was used. Amaranth flours were obtained from amaranth seeds (acquired from S.C. Solaris Plant S.R.L., Ilfov, Romania). The flours were analyzed according to the international ICC standard methods [21]: moisture content was measured according to the gravimetric method (ICC 110/1); ash content was determined in a muffle oven by incineration at 900 °C (ICC 104/1); protein content was determined with a rapid Kjeldahl device, with digestion and steam distillation (VELP Scientifica, Usmate Velate (MB), Italy), and calculated with a general factor of 6.25 for wheat flour and 5.53 for amaranth flour fractions (ICC 105/2); fat content was determined with the Soxhlet method (VELP Scientifica, Usmate Velate (MB), Italy) (ICC 136); wet gluten content and the gluten deformation index were determined according to the SR 90:2007 method [22], as a percentage of the dried substances; and total carbohydrate content was calculated by difference [13]. Salt (S.C. Sanovita S.R.L., Vâlcea, Romania) and fresh *Saccharomyces cerevisiae* yeast (S.C. Rompak S.R.L., Pascani, România) acquired from the local market were used in the bread recipe.

### 2.2. Flours Formulation and Bread Manufacturing

Amaranth seeds were milled with a laboratory Grain Mill (KitchenAid, Whirlpool Corporation, Benton Harbor, MI, USA), then sieved with a Retsch Vibratory AS 200 basic (Haan, Germany) for 30 min at 70 Hz amplitude. Three different amaranth flour (AF) particle sizes were obtained: large (L > 300 μm), medium (M > 180 μm < 300 μm), and small fractions (S < 180 μm). Amaranth flours at each particle size were added at 0, 5, 10, 15, and 20% to refined wheat flour according to the experimental design (Table 1) and mixed for 30 min in a Yucebas Y21 mixer (Izmir, Turkey). For bread manufacturing, flour blend (0.3 kg), water (at water absorption capacities of the flours previously tested on the Mixolab device, Chopin, Tripette et Renaud, Paris, France), and yeast (1.8%) were used. The bread recipe followed a biphasic method, making leaven from the quantity of water and fresh yeast, and half of the quantity of WF–AF. This mixture was left to ferment until it doubled in volume, at 30 °C and relative humidity (85%), for approximately two hours in a fermenting chamber (PL2008, Piron, Cadoneghe, Padova, Italy). The leaven obtained, and the other part of WF–AF flours with salt, were kneaded in a laboratory mixer for another 10 min (Kitchen Aid, Whirlpool Corporation, Benton Harbor, MI, USA), and left for one hour for fermenting, for each dough formulation, in the same conditions [3]. At the end of the process, the dough was cut to 400 g/piece, molded, placed in aluminum trays for one hour to produce the final fermentation, and baked for 25 min, at 220 °C (oven Caboto PF8004D, Cadoneghe, Padova, Italy).

**Table 1.** Factors and their levels in the experimental design.

| Run | A | Particle Size (μm) | B | Amaranth Flour (%) |
| --- | --- | --- | --- | --- |
| | **Coded Values** | **Real Values** | **Coded Values** | **Real Values** |
| 1 | +1.00 | 380 | 0.00 | 10 |
| 2 | −1.00 | 180 | −0.50 | 5 |
| 3 | 0.00 | 280 | −1.00 | 0 |
| 4 | 0.00 | 280 | −0.50 | 5 |
| 5 | −1.00 | 180 | −1.00 | 0 |
| 6 | +1.00 | 380 | +1.00 | 20 |
| 7 | −1.00 | 180 | 0.00 | 10 |
| 8 | +1.00 | 380 | −1.00 | 0 |
| 9 | +1.00 | 380 | −0.50 | 5 |
| 10 | +1.00 | 380 | +0.50 | 15 |
| 11 | 0.00 | 280 | 0.00 | 10 |
| 12 | 0.00 | 280 | +1.00 | 20 |
| 13 | −1.00 | 180 | +1.00 | 20 |
| 14 | 0.00 | 280 | +0.50 | 15 |
| 15 | −1.00 | 180 | +0.50 | 15 |

### 2.3. Empirical Dough Rheological Properties

The rheological behavior of the dough during the mixing and heating–cooling cycle was determined with the Mixolab equipment (Chopin, Tripette et Renaud, Paris, France) according to ICC standard method no. 173 (ICC, 2010) [21]. The mixing parameters from the registered Mixolab curves—water absorption (WA), dough development time (DT), dough stability (ST), torques related to protein weakening (C1-2), the starch gelatinization phase (C3-2), the stability of hot starch gel (C3-4), and the final starch paste viscosity after cooling at 50 °C (C5-4)—were reported.

The Falling Number index (FN) of the wheat flours and flour blends was determined by using a Falling Number device (FN 1305, Perten Instruments AB, Stockholm, Sweden) in order to determine the amylolytic activity. The dough rheological properties during extension were determined with an alveograph device (Chopin Technologies, Villeneuve-la-Garenne, France) according to ICC method no. 121 method at constant hydration to a 14% moisture basis. The determined parameters were: dough tenacity (P), dough extensibility (L), dough strength (W), and alveograph curve ratio (P/L).

The rheological properties of dough during fermentation were determined with the rheofermentometer device (Chopin Rheo, type F4, Villeneuve-La-Garenne, France) according to the AACC 89–01.01 method. The analyzed parameters were: maximum height of the gas release curve (H′m), the total volume of $CO_2$ production (VT), the volume of the gas retained in the dough at the end of the test (VR), and the retention coefficient (CR).

*2.4. Fundamental Dough Rheological Properties*

Dynamic oscillatory measurements as non-destructive tests were performed using HAAKE, MARS 40 (Thermo Scientific, Karlsruhe, Germany) with parallel plate–plates geometry. For this purpose, oscillatory frequency sweep, temperature sweep, and creep–recovery tests were performed, as reported by Iuga et al. [23]. Dough samples were preliminarily tested for the linear viscoelastic region (LVR) by applied strain sweep tests from 0.00 to 100 Pa at a constant oscillation frequency of 1 Hz. The dough, prepared at optimum water absorption capacity, but without the addition of the fresh yeast, was left to rest for 5 min before testing [23]. The excess dough was trimmed just before the measurement, and a layer of vaseline was applied to the exposed edge to avoid the evaporation of moisture during the resting period. The results, registered from a frequency sweep test applied in the LVR from 0.01 to 20 Hz, at 10 Pa stress and 20 °C, were evaluated by the elasticmodulus (G′) and the viscous modulus (G″) at 1 Hz, and the loss tangent (tan *δ*). A temperature sweep test was applied to determine the maximum gelatinization temperature (T$_{max}$), considered at the maximum G′ value by heating the dough from 20 to 100 °C at a rate of $4.0 \pm 0.1$ °C/min, a constant strain of 0.10%, and a frequency of 1 Hz.

A creep–recovery test at a constant shear stress of 25 Pa for a creep time of 60 s and a relaxation time of 180 s after removing the shear stress was applied to simulate different stresses during bread dough production [23]. Results were evaluated in terms of the maximum creep compliance (Jc$_{max}$), which corresponds to the maximum deformation at the end of the creep phase, and the maximum recovery compliance (Jr$_{max}$), associated with partial reformation after stress removal.

*2.5. Bread Physical and Textural Parameters*

After cooling, the bread samples obtained were analyzed for their bread volume (BV) according to the Romanian standard SR 90: 2007 by seed displacement method [22].

The textural parameter, bread firmness (BF), was evaluated based on the TPA mode by using a TVT-6700 texture analyzer (Perten Instruments, Hägersten, Sweden). The test speed of the probe was 1.0 mm/s. The compression strain was set at 20% while the auto-trigger force was 5.0 g, with an interval of 15 s between compressions. The firmness (BF) was recorded and processed by TexCalc 5 software (5.1.0.x. version, Perten Instruments, Hägersten, Sweden).

*2.6. Factorial Design and Statistics*

The study of PS and AF addition level effects on wheat dough rheological and bread characteristics and the optimization process were performed using Design-Expert software (Stat-Ease, Inc., Minneapolis, MN, USA). A full factorial design was used to study the main and interaction effects of two factors on 24 responses. The studied factors were three amaranth flour particle sizes (large, medium, and small) and five addition levels to wheat flour (0, 5, 10, 15, and 20%). The responses considered were the following: Falling Number (FN) index, the water absorption of the composite flour (WA), dough development time (DT), stability (DT), the consistency reached during the protein weakening stage (C1-2), the consistency reached during the starch gelatinization stage (C3-2), the consistency reached during the stability of hot starch gel (C3-4), the consistency during the starch retrogradation stage (C5-4), dough tenacity (P), dough extensibility (L), dough strength (W), alveograph configuration ratio (P/L), the maximum height of the gas release curve (H′m), the total volume of $CO_2$ production (VT), the volume of the gas retained in the dough at the end of the test (VR), the retention coefficient (CR), elastic modulus (G′), viscous modulus (G″), loss

tangent (tan δ), maximum gelatinization temperature ($T_{max}$), maximum creep compliance ($Jc_{max}$), maximum recovery compliance ($Jr_{max}$), bread volume (BV), and bread firmness (BF).

An experimental design that consists of fifteen combinations was conducted and the coded versus the real values of the factors are presented in Table 1. The simultaneous effect of these two factors on the responses was investigated through response surface methodology (RSM). The effects of PS and AF addition levels on dough and bread properties were evaluated through mathematical modeling. The most suitable model for predicting data variation for each response was selected according to *F*-test results, the coefficient of determination ($R^2$), and adjusted coefficients of determination (*Adj.*-$R^2$). The effects of the factors and their interactions were underlined using Analysis of Variance (ANOVA), considering a significance level of 95%.

In order to establish the optimal value of the factors, amaranth flour particle size, and addition level, a multiple responses analysis was applied to the predictive models in conjunction with the desirability function approach. For the numerical optimization applied in this study, the desired goal established for each response included: addition level, ST, C3-4, H'm, VT, VR, CR, W, $Jr_{max}$, and BV at maximum value, while C1-2, C5-4, P/L were minimized, and the levels of all remaining responses which are considered in this study were kept within range.

## 3. Results

### 3.1. Flour Chemical Characteristics

The values of the chemical characteristics of the wheat flour and amaranth particle sizes, large (AL), medium (AM), and small (AS), are presented in Table 2. The wheat flour studied presented a low α amylase activity, 30.00% wet gluten content, and 6.00 mm gluten deformation index, characteristics that made it suitable for bread making according to Romanian standard SR 90:2007 [22].

**Table 2.** One-way analysis of variance (ANOVA) of the chemical composition of the wheat flour in comparison with amaranth flour fractions: large particle size (AL), medium particle size (AM) and, small particle size (AS).

| Parameters | Wheat Flour | Amaranth Flour Particle Size | | |
| --- | --- | --- | --- | --- |
| | | **AL** | **AM** | **AS** |
| Moisture (%) | 14.08 ± 0.08 [a] | 10.61 ± 0.05 [b] | 10.19 ± 0.06 [c] | 9.35 ± 0.06 [d] |
| Ash (%) | 0.69 ± 0.05 [e] | 1.62 ± 0.02 [c] | 3.54 ± 0.04 [b] | 4.45 ± 0.04 [a] |
| Protein (%) | 12.45 ± 0.15 [c] | 10.18 ± 0.44 [d] | 25.33 ± 0.25 [b] | 29.36 ± 0.01 [a] |
| Fat (%) | 1.41 ± 0.01 [d] | 7.49 ± 0.02 [b] | 8.09 ± 0.04 [a] | 7.11 ± 0.01 [c] |
| Carbohydrates (%) | 71.36 ± 0.02 [a] | 71.16 ± 0.06 [a] | 52.85 ± 0.53 [b] | 49.74 ± 0.09 [c] |
| FN | 312 ± 3.12 | - | - | - |
| WGC | 30.00 ± 0.15 | - | - | - |
| GDI | 6.00 ± 0.12 | - | - | - |

AL—amaranth large particle size; AM—medium particle size; AS—small particle size. FN—Falling Number index; WGC—wet gluten content; GDI—gluten deformation index. Lower-case letters ([a–e]) refer to the comparison of the same compound between the different particle size amaranth flour samples; results followed by the lowercase letter are significantly different according to Tukey's HSD post hoc test ($p < 0.05$).

A decrease in sample moisture can be observed as the particle size decreased, being significantly ($p < 0.05$) lower than wheat flour. The ash content varied between 1.62 and 4.45%, which was significantly higher than ash from wheat flour (0.69%), which increased when the particle size decreased, being significantly different between samples. The protein content from amaranth fractions ranged between 10.18 and 29.36% and followed the same upward trend, as did the ash content, and was significantly higher than wheat flour (12.45%). The fat content of amaranth particle size flours ranged between 7.11 and

8.09%, the highest values were observed in medium particle size (AM), with all samples showing significant ($p < 0.05$) differences between them, and all fractions being significantly higher than the fat content from wheat flour (1.41%). The total carbohydrate content for the amaranth particle fractions presented significant ($p < 0.05$) differences between all samples, the highest content being at large fractions, while the lowest carbohydrate content was present in small fractions (AS).

### 3.2. Influence of Particle Size and Addition Level of Amaranth Flour to Wheat Flour on FN Index, Dough Rheological Properties during Mixing and Heating-Cooling Cycle, and Dough Biaxial Extension

The accuracy test of the model (Table 3) showed that the quadratic and 2FI models properly predict the studied parameters as a function of the formulation factors. The data for the Falling Number index, Mixolab, and alveographic parameters were successfully fitted ($p < 0.05$) to the quadratic model, which explained proportions of 66–96% of the variation data, as the ANOVA results showed (Table 3). The 2FI mathematical model chosen for dough development time (DT) and the consistency reached during the starch gelatinization stage (C3-2) data fitting explained 68 and 95%, respectively, of the variation, and it was significant at $p < 0.05$ in both cases.

**Table 3.** The coefficients in the predictive models for the FN index, Mixolab, and alveograph parameters.

| | | | | | Parameters | | | | | | | |
|---|---|---|---|---|---|---|---|---|---|---|---|---|
| Factors | Falling Number | | | | Mixolab | | | | | Alveograph | | |
| | FN (s) | WA (%) | DT (min) | ST (min) | C1-2 (Nm) | C3-2 (Nm) | C3-4 (Nm) | C5-4 (Nm) | P (mm) | L (mm) | W ($10^{-4}$ J) | P/L |
| Constant | 317.06 | 58.65 | 2.98 | 9.65 | 0.61 | 1.23 | 0.15 | 0.80 | 95.77 | 48.53 | 160.70 | 1.87 |
| A | −1.96 | −0.52 ** | 0.45 | 1.17 ** | −0.05 ** | 0.05 ** | 0.08 ** | −0.00 | −6.38 | 1.88 | 6.08 | −0.17 |
| B | −2.98 | 0.39 ** | 1.53 ** | −1.06 ** | 0.04 ** | −0.15 *** | 0.12 *** | −0.27 *** | 9.62 ** | −28.55 *** | −50.94 ** | 1.38 *** |
| A × B | −5.85 * | −0.77 ** | 0.47 | 0.77 * | −0.04 ** | 0.03 * | 0.07 ** | −0.03 | −4.50 | 2.50 | −5.30 | −0.43 |
| A$^2$ | −2.40 | −0.42 * | | −0.55 | 0.00 | - | −0.02 | −0.03 | 2.40 | −2.30 | −5.30 | 0.08 |
| B$^2$ | −7.90 * | 0.58 * | | −0.52 | 0.03 * | - | 0.03 | 0.07 | 1.33 | 13.43 ** | 33.90 * | 0.29 |
| $R^2$ | 0.71 | 0.93 | 0.68 | 0.83 | 0.93 | 0.95 | 0.96 | 0.89 | 0.69 | 0.96 | 0.85 | 0.86 |
| *Adj.-$R^2$* | 0.55 | 0.89 | 0.59 | 0.74 | 0.90 | 0.93 | 0.94 | 0.82 | 0.51 | 0.93 | 0.76 | 0.79 |
| *p-value* | 0.0265 | <0.0001 | 0.0047 | 0.0028 | <0.0001 | <0.0001 | <0.0001 | 0.0005 | 0.0352 | <0.0001 | 0.0018 | 0.0011 |

\*\*\* $p < 0.001$; \*\* $p < 0.01$; \* $p < 0.05$; A—particle size (μm); B—level of amaranth flour added to refined wheat flour (%); $R^2$, *Adj.-$R^2$*—measures of model fit; FN—Falling Number; WA—water absorption; DT—development time; ST—stability; C1-2—consistency reached during protein weakening stage; C3-2—consistency reached during starch gelatinization stage; C3-4—consistency reached during the stability of hot starch gel; C5-4—consistency during starch retrogradation stage; P—dough tenacity; L—dough extensibility; W—deformation energy; P/L—alveograph curve ratio.

The ANOVA for the quadratic model, as fitted to the experimental results, showed significance ($p < 0.05$). The FN index ranged from 289 to 321 s for the composite flours' formulation. The FN was significantly correlated ($p < 0.05$) with the interaction effect of the AF addition level to WF and particle size and, also, with the quadratic effect of the AF addition level, in a negative way. The effects of particle sizes and AF addition level is shown in Figure 1a, indicating a decrease in the FN index with an increase in the AF addition and PS.

During mixing, the dough was influenced by PS and AF addition to wheat flour. Water absorption registered a significant ($p < 0.05$) decrease (Figure 1b) when the PS increased and the AF addition level decreased, while the interaction between factors had a significant ($p < 0.05$) negative influence (Table 3). The dough development time (DT) showed a significant ($p < 0.05$) increase (Figure 1c) when the AF addition level was increased, while the PS had a non-significant effect ($p > 0.05$), which ranged between 1.33 and 5.75 min. The dough stability (ST) was significantly ($p < 0.05$) negatively affected by the AF addition level (Table 3), while the PS and the interaction between the factors had a positive significant influence on the ST, which ranged between 5.60 and 10.53 min. The rise in AF amounts led to proportionally lower dough stability (Figure 1d).

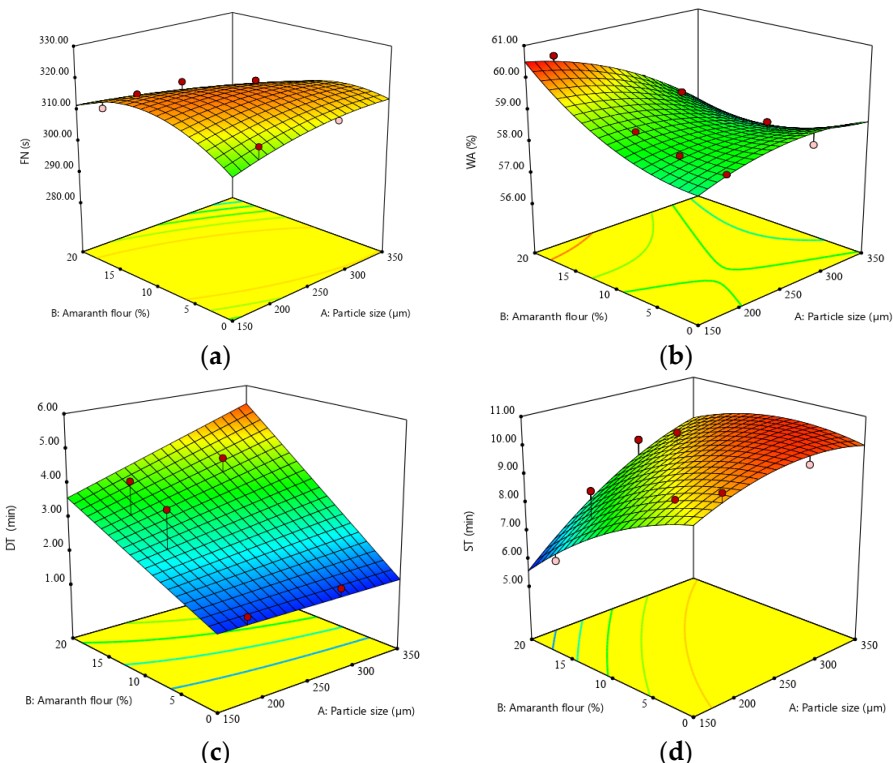

**Figure 1.** Three-dimensional response surface plots showing the interaction between amaranth flour particle size and addition level on (**a**) the Falling Number index (FN), (**b**) water absorption (WA), (**c**) development time (DT), and the (**d**) dough stability (ST) achieved during mixing.

Protein weakening (C1-2) values decreased significantly ($p < 0.05$) when the PS increased (Figure 2a), while this parameter was raised when the AF addition level was increased. The interaction between factors significantly affected protein weakening (C1-2 torque), indicating a decrease in protein weakening speed under the effect of temperature increase. The starch gelatinization stage showed a significant ($p < 0.05$) decrease as the addition level of AF was increased, while the PS and the interaction between factors significantly positively influenced C3-2. The effect of particle size and the AF addition level to wheat flour can be seen in Figure 2b, indicating an increase in C3-2 with particle size increase. The stability of hot starch gel (C3-4) was significantly positively affected by both factors and by the interaction between them. The effect of particle size and addition level on C3-4 is presented in Figure 2c, showing the capacity of the AF fractions, the addition level, and their interaction to increase the C3-4. The starch retrogradation during cooling was significantly ($p < 0.05$) decreased with AF addition (Figure 2d), while particle size and the interaction between factors showed a non-significant influence on C5-4 ($p > 0.05$).

The effects of factors on dough extension properties are presented in Figure 3. It can be seen from Table 3 that dough tenacity was significantly ($p < 0.05$) affected by the AF addition level, while particle size showed a non-significant effect ($p > 0.05$) on this parameter. It can be observed in Figure 3a that when the AF amount was increased, the dough tenacity was also increased. At the same time, when the dough tenacity increased with AF addition, the dough extensibility decreased significantly ($p < 0.05$) with an increase in this factor. The quadratic term of AF influences in a positive way the extensibility (Figure 3b). The quadratic regression model was fitted for the dough strength, which indicates that this alveograph parameter was significantly ($p < 0.05$) influenced by the linear and quadratic term of the AF addition level. It can be observed in Figure 3c that when the AF amount increased, the dough strength decreased. The alveograph configuration ratio presented a significant ($p < 0.05$) increase only with AF addition level increase (Figure 3d).

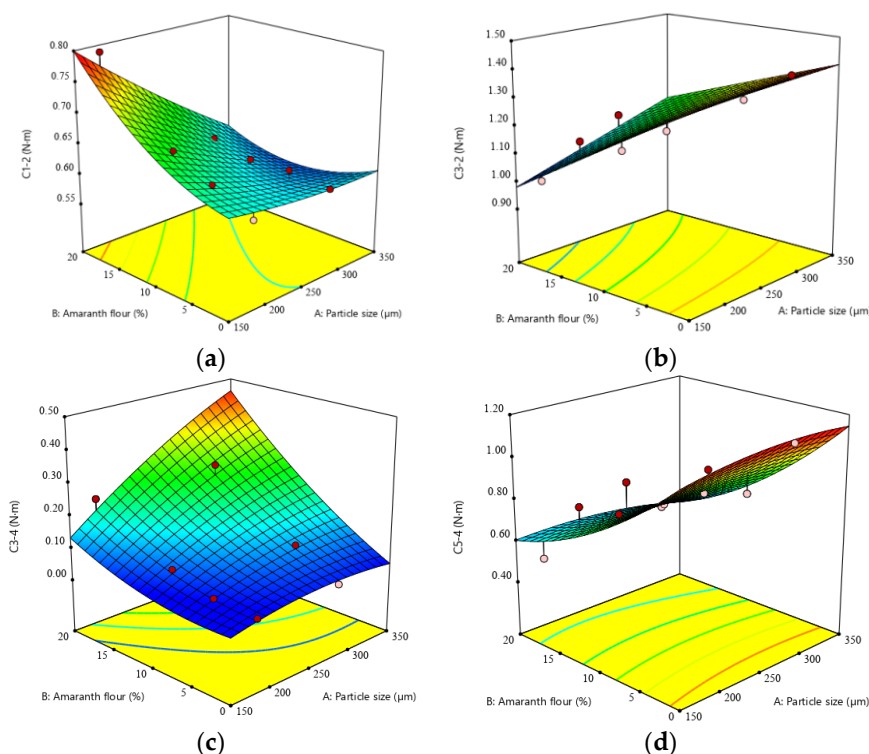

**Figure 2.** Three-dimensional response surface plots showing the interaction between amaranth flour particle size and addition level on: (**a**) protein weakening (C1-2; (**b**) starch gelatinization (C3-2); (**c**) the stability of hot starch gel (C3-4); and (**d**) starch retrogradation (C5-4).

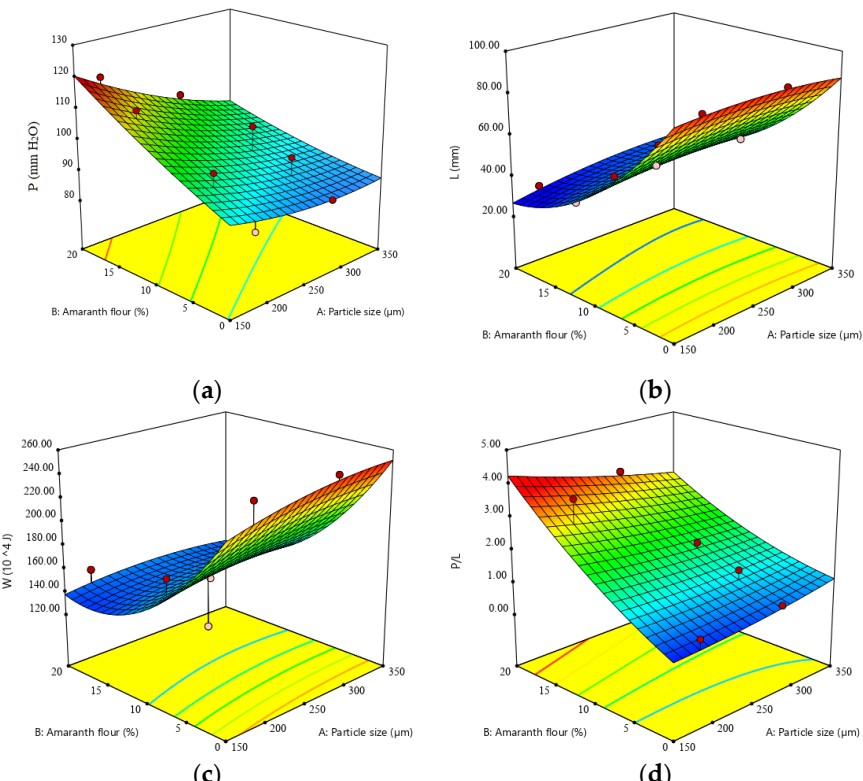

**Figure 3.** Three-dimensional response surface plots showing the interaction between amaranth flour particle size and addition level on: (**a**) dough tenacity (P); (**b**) dough extensibility (L); (**c**) dough strength (W); and (**d**) alveograph configuration ratio (P/L).

### 3.3. Influence of Particle Size and Addition Level of Amaranth Flour in Wheat Flour on Dough Fermentation, Dynamic Rheological Properties, and Bread Characteristics

The quadratic model successfully fitted ($p < 0.05$) the data for the maximum height of the gas release curve (H′m), the volume of gas retained in the dough at the end of the test (VR), the retention coefficient (CR), elastic modulus (G′), loss tangent (tan δ), maximum gelatinization temperature ($T_{max}$), maximum creep compliance ($Jc_{max}$), maximum recovery compliance ($Jr_{max}$), bread volume (BV), and bread firmness (BF). The variations were explained in proportions of 66 to 92% (Table 4). For total $CO_2$ volume production (VT) and viscous modulus (G″) data prediction, the 2FI model was found to be adequate ($p < 0.05$), with an explained variation of 70 and 71%, respectively (Table 4).

**Table 4.** The coefficients in the predictive models for dough during fermentation, dynamic rheological properties, and bread properties.

| Factors | Parameters | | | | | | | | | | | |
|---|---|---|---|---|---|---|---|---|---|---|---|---|
| | Rheofermentometer | | | | Rheometer | | | | | | Bread Parameters | |
| | H′m (mL) | VT (mL) | VR (mL) | CR (%) | G′ (Pa) | G″ (Pa) | tan δ (adim.) | $T_{max}$ (°C) | $Jc_{max}$ (Pa$^{-1}$) | $Jr_{max}$ (Pa$^{-1}$) | BV (cm$^3$) | BF (g) |
| Constant | 73.61 | 1268.45 | 1180.87 | 91.53 | 28,093.97 | 10,500.65 | 0.3476 | 79.32 | 20.27 | 13.40 | 360.83 | 951.01 |
| A | 0.25 | 35.40 | 16.96 | −0.39 | 3939.00 | 1632.50 ** | 0.0144 | −0.14 | −7.52 ** | −4.42 ** | 26.26 ** | −365.40 * |
| B | 6.30 ** | 84.58 ** | 89.36 ** | 1.20 | 8848.23 ** | 1463.54 * | −0.0375 ** | −1.74 ** | −1.46 | −0.87 | −40.73 *** | 673.85 ** |
| A × B | 1.40 | 29.40 | 8.80 | −1.15 | 2667.00 | 1837.65 * | 0.0217 | −0.57 | −1.71 | −1.59 | 7.70 | 202.60 |
| A$^2$ | 0.44 | - | −14.10 | −0.99 | −1172.50 | - | −0.0205 | 0.52 | 3.68 | 1.77 | −30.11 ** | 602.50 ** |
| B$^2$ | −4.53 | - | −76.57 ** | −4.36 ** | 9655.71 * | - | 0.0275 | 1.67 | 2.14 | 1.70 | −6.30 | 195.57 |
| $R^2$ | 0.80 | 0.70 | 0.87 | 0.80 | 0.81 | 0.71 | 0.72 | 0.85 | 0.66 | 0.71 | 0.92 | 0.88 |
| *Adj.-R$^2$* | 0.68 | 0.62 | 0.80 | 0.69 | 0.71 | 0.64 | 0.57 | 0.77 | 0.47 | 0.55 | 0.87 | 0.81 |
| *p-value* | 0.0062 | 0.0032 | 0.0009 | 0.0054 | 0.0041 | 0.0025 | 0.0221 | 0.0015 | 0.0492 | 0.0268 | 0.0001 | 0.0007 |

*** $p < 0.001$; ** $p < 0.01$; * $p < 0.05$; A—particle size (μm); B—level of amaranth flour added to refined wheat flour (%); $R^2$, *Adj.-R$^2$*—measures of model fit; H′m—maximum height of the gas release curve; VT—total $CO_2$ volume production; VR—volume of the gas retained in the dough at the end of the assay; CR—retention coefficient; G′—elastic modulus; G″—viscous modulus; tan δ—loss tangent; $T_{max}$—maximum gelatinization temperature; $Jc_{max}$—maximum creep compliance; $Jr_{max}$—maximum recovery compliance; BV—bread volume; BF—bread firmness.

Table 4 shows the effects of PS and AF addition formulation factors on the maximum height of the gas release curve (H′m). The regression model indicates that the linear term of the addition level had a significant ($p < 0.05$) influence on the H′m parameter, while the PS had a non-significant effect ($p > 0.05$).

A response surface plot, showing the effect of AF level and PS on H′m, is represented in Figure 4a, and it can be seen that the H′m significantly increased as the AF level increased. The effect of the particle size and AF addition level on the total $CO_2$ volume production of amaranth–wheat composite flour dough as their corresponding regression coefficients in the 2FI model indicated is presented in Table 4. The VT varied from 1123 mL to 1421 mL, which was lower compared to the wheat control (1168 mL), but an increase in VT can be observed with the increase of AF addition, while the PS had no significant ($p > 0.05$) influence (Figure 4b). The volume of the gas retained in the dough at the end of the test (VR) was influenced significantly ($p < 0.05$) by the linear term of the AF addition level and the quadratic term of this factor (Table 4). The regression model for the VR (Table 4) showed a non-significant ($p > 0.05$) effect in linear terms of particle size and in terms of the interaction between particle size and AF addition level. The AF addition level positively affects VR, while the quadratic term of the AF addition level negatively influences it (Figure 4c). The effect of the PS and AF level on the CR, expressed as their corresponding regression coefficients in the quadratic regression model, is shown in Table 4. The CR was significantly influenced ($p < 0.05$) by the quadratic term of the AF level in a negative way. The response surface obtained for the CR (Figure 4d) showed that the increase in AF level decreased the CR parameter.

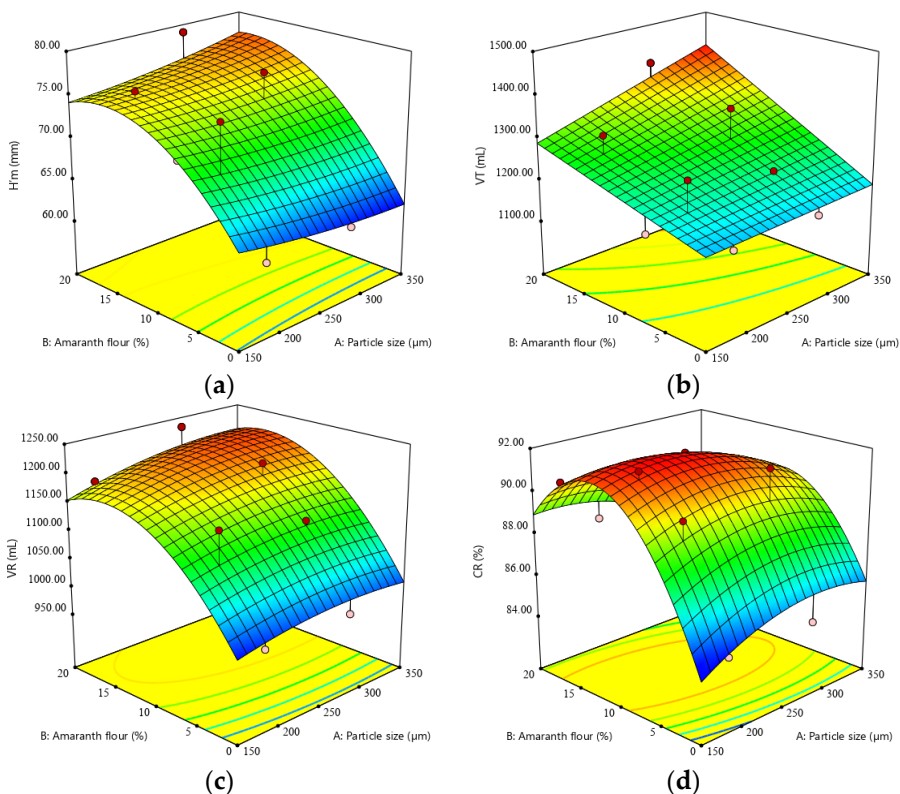

**Figure 4.** Three-dimensional response surface plots showing the interaction between amaranth flour particle size and addition level on: (**a**) maximum height of the gas release curve (H′m); (**b**) total $CO_2$ volume production (VT); (**c**) volume of the gas retained in the dough at the end of the test (VR); and the (**d**) retention coefficient (CR).

The dynamic dough rheological properties indicated different trends depending on particle sizes and AF addition levels. The elastic modulus (G′) was significantly influenced ($p < 0.05$) by the level of AF addition to WF, while the particle size had no significant influence ($p > 0.05$) (Table 4). G′ was influenced significantly ($p < 0.05$) by the linear term of the AF and the quadratic term of the AF addition level (Figure 5a), while the PS and the interactions between them presented a non-significant ($p > 0.05$) influence. The 2FI predictive model results from the regression analysis showed that both the PS and the addition level of AF and their interaction were significantly ($p < 0.05$) influenced and fitted well with the experimental data for G″ (Table 4). An increasing trend was observed with the increase in PS and AF addition levels (Figure 5b). The loss tangent (tan δ) decreased significantly ($p < 0.05$) as the amount of AF was increased (Figure 5c). Through ANOVA, the quadratic model was found to adequately represent the experimental data for the maximum gelatinization temperature ($T_{max}$), the $R^2$ value (0.85), confirming the adequacy of the model. The linear coefficient of the addition level indicated a significant ($p < 0.05$) negative influence on the $T_{max}$, while the linear term of the PS and the interaction between factors were not significant. The effect of the PS and AF addition level can be seen in Figure 5d, indicating a decrease in $T_{max}$ with the increase in AF.

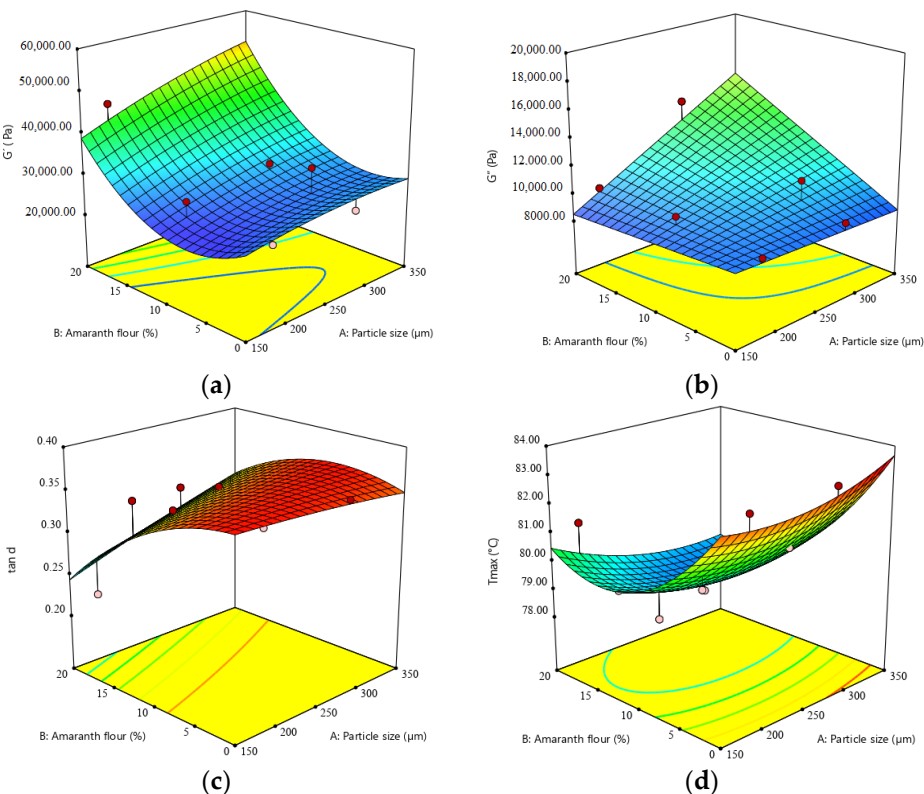

**Figure 5.** Three-dimensional response surface plots showing the interaction between amaranth flour particle size and addition level on: (**a**) elastic modulus (G′); (**b**) viscous modulus (G″); (**c**) loss tangent (tan δ); and (**d**) maximum gelatinization temperature ($T_{max}$).

The maximum creep compliance ($Jc_{max}$) was significantly influenced by the PS of amaranth flour, while the addition level did not influence this dynamic rheological parameter (Figure 6a). The recovery phase compliance was influenced ($p < 0.05$) by the particle size levels, while the AF added to wheat flour had a non-significant influence ($p > 0.05$). It was found that the PS had a highly significant negative effect ($p < 0.01$) on $Jr_{max}$, showing a decrease in $Jr_{max}$ with the increase in PS (Figure 6b).

Bread firmness is an essential parameter in determining product quality, which determines its shelf-life. The crumb firmness was significantly influenced ($p < 0.05$) by the linear term of the PS, AF addition level, and the quadratic term of the PS. The response surface was generated (Figure 6c) to predict the bread firmness as a simultaneous function of the PS and AF added to wheat flour, and this physical parameter decreased significantly when the PS was increased. Otherwise, the bread firmness increased with greater AF amounts. The bread volume (BV) followed an inverse trend, being significantly positively affected by the PS, and negatively affected by the AF added to wheat flour. The results showed an increase in BV when the PS was higher, but it can be observed that this parameter decreased with the increase in the AF addition level (Figure 6d).

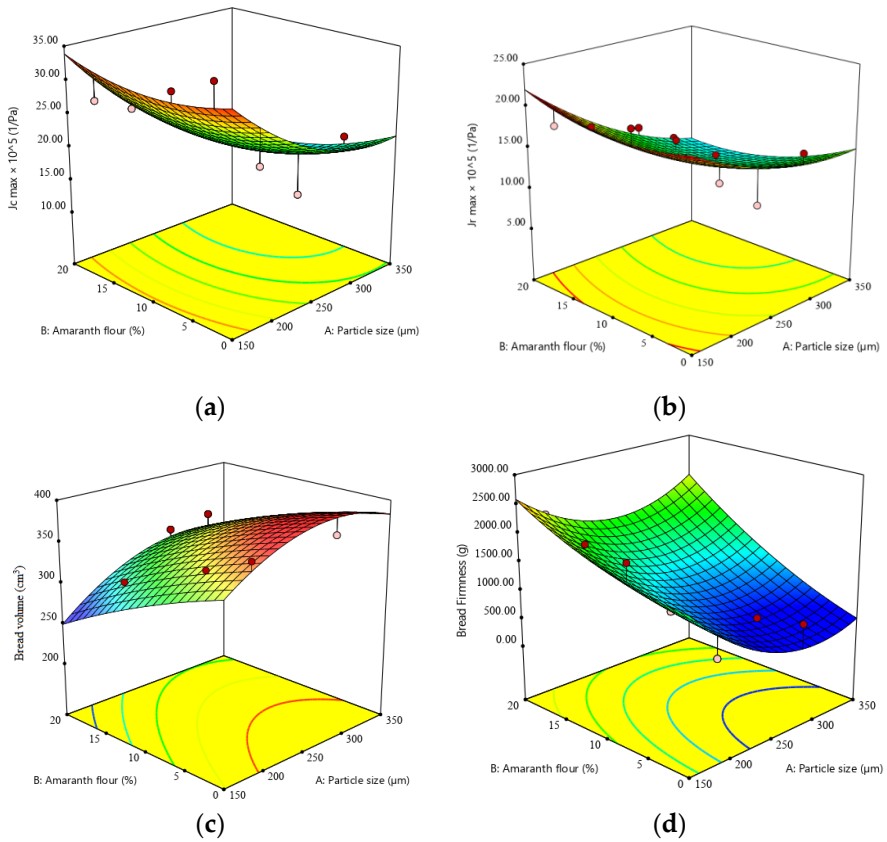

**Figure 6.** Three-dimensional response surface plots showing the interaction between amaranth flour particle size and addition level on: (**a**) maximum creep compliance ($Jc_{max}$); (**b**) maximum recovery compliance ($Jr_{max}$); (**c**) bread volume (BV); and (**d**) bread firmness (BF).

### 3.4. Optimal and Control Samples Properties

The optimal addition levels for each PS and the predicted values of the responses are presented in Table 5.

**Table 5.** Wheat flour dough and optimized composite flour for each amaranth flour particle size.

| Parameters | Control Sample | O_AL | O_AM | O_AS |
|:---:|:---:|:---:|:---:|:---:|
| **Addition Level** | **100% WF** | **9.41%** | **9.39%** | **7.89%** |
| FN (s) | 312.00 | 311.00 | 316.50 | 316.66 |
| WA (%) | 58.50 | 57.28 | 58.44 | 58.63 |
| DT (min) | 1.69 | 3.44 | 3.01 | 2.41 |
| ST (min) | 9.96 | 10.23 | 9.99 | 8.87 |
| C1-2 (Nm) | 0.61 | 0.56 | 0.60 | 0.64 |
| C3-2 (Nm) | 1.41 | 1.30 | 1.25 | 1.23 |
| C3-4 (Nm) | 0.05 | 0.22 | 0.17 | 0.07 |
| C5-4 (Nm) | 1.15 | 0.76 | 0.82 | 0.85 |
| P (mm $H_2O$) | 87.00 | 90.73 | 93.44 | 99.10 |
| L (mm) | 91.00 | 48.51 | 50.60 | 53.05 |
| W $\times 10^{-4}$ (J) | 253.00 | 162.92 | 165.33 | 165.28 |
| P/L (adim.) | 0.95 | 1.74 | 1.75 | 1.69 |
| H′m (mm) | 62.00 | 74.22 | 73.30 | 72.32 |
| VT (mL) | 1168.00 | 1307.73 | 1273.44 | 1230.20 |
| VR (mL) | 991.00 | 1173.33 | 1178.87 | 1141.19 |
| CR (%) | 84.80 | 89.34 | 91.25 | 90.70 |

**Table 5.** *Cont.*

| Parameters | Control Sample | O_AL | O_AM | O_AS |
|:---:|:---:|:---:|:---:|:---:|
| **Addition Level** | **100% WF** | **9.41%** | **9.39%** | **7.89%** |
| $G'$ (Pa) | 26,370.00 | 30,587.03 | 28,624.29 | 23,720.87 |
| $G''$ (Pa) | 9488.00 | 12,411.70 | 10,869.10 | 9320.63 |
| tan δ (adim.) | 0.3600 | 0.3630 | 0.3550 | 0.3460 |
| $T_{max}$ (°C) | 83.24 | 80.15 | 79.44 | 80.02 |
| $Jc_{max} \times 10^{-5}$ (1/Pa) | 24.50 | 16.91 | 18.47 | 27.48 |
| $Jr_{max} \times 10^{-5}$ (1/Pa) | 16.62 | 10.81 | 12.32 | 17.38 |
| BV (cm$^3$) | 372.20 | 345.73 | 368.28 | 337.11 |
| BF (g) | 786.00 | 1443.42 | 852.09 | 1398.75 |

O_AL—optimized samples with amaranth large particle size; O_AM—optimized samples with amaranth medium particle size; O_AS—optimized samples with amaranth small particle size; FN—Falling Number; WA—water absorption; DT—dough development time; ST—dough stability; C1-2—consistency reached during protein weakening stage; C3-2—consistency reached during starch gelatinization stage; C3-4—consistency reached during the stability of hot starch gel; C5-4—consistency during starch retrogradation stage; P—dough tenacity; L—dough extensibility; W—deformation energy; P/L—configuration ratio of the alveograph curve; H′m—maximum height of the gas release curve; VT—total $CO_2$ volume production, VR—volume of the gas retained in the dough at the end of the test; CR—retention coefficient; $G'$—elastic modulus; $G''$—viscous modulus; tan δ—loss tangent; $T_{max}$—maximum gelatinization temperature; $Jc_{max}$—maximum creep compliance; $Jr_{max}$—maximum recovery compliance; BV—bread volume; BF—bread firmness.

## 4. Discussion

### 4.1. Proximate Composition of Amaranth Particle Sizes

The higher shear force which was applied to the milling process can explain the lowest moisture in the finest amaranth particle flour. The biggest content of ash and protein content were in the small fractions, followed by medium PS, a fact that can be linked to the location of the protein in the germ (65%), and 35% in the endosperm of amaranth seed, compared to an average of 15 and 85%, respectively, in most cereals [24,25]. The protein content from the finest amaranth fractions is comparatively higher than the protein content from the same size of quinoa and buckwheat fractions [3]. Amaranth fractions contain higher lipids than most cereals [25].

### 4.2. Influence of PS and AF Addition Level on Falling Number, Mixolab, and Alveographic Parameters

The FN value is inversely correlated with α-amylase activity, so it may be concluded that with the increase in the AF addition, α-amylase activity increases. This increase can be correlated with the presence of calcium ions in amaranth [14,26,27]. High α-amylase activity is desired for improving the final product quality. The increase in WA with the increase in AF particle size and addition level may be explained by the gluten dilution, which requires less hydration, and therefore a lower amount of water is needed [28]. On the other hand, an increase in WA can be observed when the particle size decreases. This trend can be explained by the specific surface area of amaranth starch, which is larger than wheat starches, which can absorb more water compared to large and medium particle fractions. A similar result was reported by Iuga et al., Mironeasa et al., and Ahmed et al. [23,29,30], whereby small particle sizes of non-gluten flour increased the values of WA. Dough development time measures the dough strength and was significantly affected by the PS, demonstrating that large particles require a long time to reach the optimal elastic and viscosity characteristics. The addition of non-gluten flour affects gluten quality with an increased degree of softening, which is reflected in a higher DT and a lower ST. This behavior can be explained by the presence of larger amounts of crude fiber in the amaranth flour that tend to imbibe and retain water in the dough, affecting the mixing process parameters. Moreover, a longer development time could be linked with the difference in the rate of water absorption by wheat and amaranth flours, due to higher amounts of soluble proteins in amaranth flour and maybe also due to the water absorption characteristics of amaranth starch and non-starch polysaccharides. This may cause a delayed formation of the gluten network

in the dough [31]. Otherwise, when the PS increases, the DT is improved, this fact being associated with amaranth albumins that interact with gluten proteins through disulfide bonds [31,32].

The protein weakening (C1-2) increased when the PS decreased, which can be explained by the changes in protein network structure, being more available for enzymatic attacking points and leading to a rise in the speed of protein weakening due to the heat [33]. Excessive mechanical impact weakens dough consistency, which could probably be explained by the high shear temperature from the milling process, which leads to water-binding by protein substances. Another factor might be the better retention of the hydration shell on the protein globules [34]. Starch gelatinization is a key factor in starch behavior, which occurs when the dough is heated at 60 °C. Lower values of C3-2 can be explained by the increase in the interactions between the low amount of amylose (1–5%) and higher long-chain amylopectin (20–25%) from amaranth starch, which generate a synergistic effect on the final viscosity and, thus, on starch retrogradation [35,36]. The combined effect of starch hydrolysis and a low FN index will lead to a decrease in viscosity [37], which will increase the water amount from the dough [38,39]. An increase in hot gel stability values may be related to the starch damage process. Our results indicate that amaranth flour addition can limit starch retrogradation, increasing bread freshness and shelf-life. The biaxial extension of the dough can be monitored with an alveograph analysis, which monitors vital parameters that can improve the fermentation process.

The dough tenacity (P), characterized by the force required for dough rupture, can predict the bread volume [40,41]. This alveographic parameter increased when the AF addition level increased, a fact that could be explained by the non-gluten flour incorporation, or could be a consequence of the higher level of protein and fiber from amaranth seeds [42,43]. The extensibility of dough (L) is the property of wheat flour dough to obtain the characteristic structure and volume of baked goods [44], can predict the handling properties of the dough [45], and was greatly reduced by the AF addition. This phenomenon can be explained by the small numbers of hydroxyl groups from fiber which has more water availability and the weakening of the gluten network [46]. Similar data were reported by Piga et al. [36] when amaranth flour was studied as a potential healthy ingredient for the development of an innovative gluten-free flatbread. Dough strength or deformation energy (W) decreased significantly when AF amounts were increased, while the PS did not influence this parameter, possibly due to the differences in protein content between wheat flour and amaranth flour [43]. The P/L ratio, which gives information about the elastic resistance and extensibility balance of dough, was augmented in doughs with amaranth flour.

*4.3. Influence of PS and AF Addition Levels on Dough Fermentation, Dynamic Rheological Properties, and Bread Characteristics*

The maximum height of the gas release curve (H'm) is a critical parameter in the fermentation process and is related to the maximum height of dough development and the height of dough development at the end of the test. The H'm parameter was closely related to the bread volume [47], and therefore it is a good attribute for predicting the final product. The dough, which contains non-diluting gluten, will be more elastic and will form bread with a continuous sponge structure after baking [48]. The gas retention was decreased, which may be due to dough permeability when the gluten network was weakened by the amylose and amylopectin hydrolysis on the presence of enzymes during the fermentation process. Martínez and Gómez (2017) [49] reported that the structure and morphology of the starch granules and flour particles were the major determinants of the dough changes produced during the fermentation and baking phases. Some authors reported that the greater milling damage to the starch level in superfine-ground flour gave rise to a decrease in total gas production during dough fermentation [50] due to the amylase inactivation, thus hindering the formation of fermentable sugars and reducing the capacity of yeasts to produce gas [51]. The compact particles of the flours produced dough with high consistency and bread with volume and textural properties lower than

those obtained with large particles. These changes might be due to the competitiveness of water uptake between wheat gluten and AF components such as starch, protein, and fiber. The highest swelling capacity value was found for the small AF particle size due to an increase in damaged starch, which influences dough behavior [3].

Fundamental rheological tests can offer valuable information about the final product. An increase in elastic (G′) and viscous (G″) moduli values could be linked to the presence of binding agents from the composite dough and attractive forces between starch granules and amaranth fibers, this trend suggesting a solid, elastic-like behavior of the WF–AF composite dough. The higher values of G′ and G″ could probably also be a result of interactions between proteins and the formation of a protein–starch complex. The significant increase in G′ with the increased addition level of AF to wheat flour could be related to the increasing amounts of damaged starch in the composite flour. Hatcher et al. [52] found that the damaged starch in wheat flour greater affected the viscoelastic properties of noodles than the particle size, resulting in stiffer doughs, than did flour with low starch damage. Decreases in loss tangent (tan δ) values are typical for elastic and firm doughs. Similar results were found by Burešová et al. [53]. The tan δ values are influenced by the level of starch damage. The dough from flours with high starch damage presents significantly lower values than those with low or medium starch damage [52].

A decrease in the maximum gelatinization temperature with the increase in the AF addition could be due to the higher amount of water absorbed by the amaranth grain, and the greater swelling power and solubility of the amaranth starch granule compared with the wheat starch granule [54]. A higher amount of water absorbed could be related to the starch damage from the amaranth flour milling process.

The maximum creep compliance (Jc_max) and the maximum recovery compliance (Jr_max), measured at the end of the creep and recovery phases, respectively, represent the principal characteristics from the creep–recovery curves. The results of the creep–recovery measurements are significantly influenced by the protein or starch levels from amaranth fractions, while the AF addition level did not significantly affect these dynamic parameters. This strengthening phenomenon can be related to the hydroxyl groups from sugar compounds from amaranth seeds, which may directly interact with proteins, resulting in non-covalent or covalent bonding. Protein–polyphenol interactions modify proteins, influencing the quality and functional properties of a food [23,55]. The bread volume decreased as the AF addition level was increased, which could be explained by the poor baking quality of this flour, due to its lack of gluten proteins, which are present in wheat [25]. The proteins present in amaranth flour consist of three major fractions, albumins (51%), globulins (16%), and glutelins (24%), and a minor, alcohol-soluble fraction, prolamine, between 1.4 and 2.0% [56,57]. The albumin fraction is comparable with egg-white proteins and can be used as an egg substitute in different products [25]. The decrease in bread volume with the AF addition can be linked to the weakening of the gluten matrix and reduced gas retention of the dough (Table 4), which is predictable for the higher dough tenacity and lower extensibility (Table 3). Our results fall in line with those of Tömösközi et al. [31]. The high lipid content from small and medium amaranth fractions (Table 2) could have functionality as a gas stabilizing agent during breadmaking, which probably improves the bread's technological properties based on medium and small amaranth fractions (volume, elasticity) [58]. Some authors found a direct relation between dough elasticity/crumb chewiness and crumb firmness [59]. These results are in agreement with those reported by other authors [27,60–62].

The amaranth flour addition level negatively influenced the bread firmness; an increase in bread firmness was observed when the AF amount was increased. Regarding amaranth particle sizes, there was a decrease in bread firmness as the particle size was increased (especially for the 5–10% addition level), a phenomenon that can be explained by the albumin proteins from amaranth grain, which can act like gluten in the dough and have the capacity to interact with wheat glutenin protein through disulfide bonds, which does not weaken the gluten network overmuch. Similar results were obtained by Oszvald et al. and Miranda Ramos et al. [32,61].

*4.4. Optimal PS and Addition Levels of AF*

The numerical optimization procedure revealed that the most suitable composite flour, based on the large amaranth particle size (380 μm), would have 9.41% amaranth flour and 90.59% wheat flour; for the medium amaranth particle size (280 μm), a percent of 9.39% amaranth flour and 90.61% wheat flour would be most suitable; and for the small amaranth particle size (180 μm), the most suitable amaranth flour addition level is 7.89% to 92.11% wheat flour. According to the obtained results, both optimal and control flours present dough stability (ST) and protein weakening (C1-2) very close values. The development time (DT) was considerably higher than the control sample. In this case, the optimal dough formulations can retain gas easier than the wheat dough. The differences in the Falling Number, empirical, and dynamic rheological properties, and bread physical and textural properties between the optimal and control samples can be linked to the significance of proteins, lipids, carbohydrates, and minerals of the amaranth fractions and their interactions in the dough network.

## 5. Conclusions

This research studied the influence of amaranth flour particle size and addition level on the wheat dough rheological, technological, and textural bread properties, to optimize dough and bread quality in baking. Small fractions, followed by medium ones, presented the highest content of protein and ash, while the large particle sizes have a high content of carbohydrates.

The results obtained revealed that particle size and AF addition level modified dough behavior during mixing, extension, and fermentation, as well as the bread's physical and textural parameters. Dough rheology and bread parameters were significantly influenced by particle size and the addition level of amaranth flour, all the regression models obtained for responses being significant ($p < 0.05$) and with high coefficients of determination ($R^2 = 0.66$–$0.96$).

The combined effect of AF particle size and addition level on wheat flour led to a decrease in the Falling Number, starch gelatinization, the final starch paste viscosity after cooling, dough extensibility, baking strength, maximum gelatinization temperature, maximum creep and recovery compliance, and bread volume, while dough development and stability, protein weakening, the stability of hot starch gel, dough tenacity, the alveograph curve ratio, all rheofermentometer parameters (maximum height of the gas release curve, the total volume of $CO_2$ production, the volume of the gas retained in the dough at the end of the test, the retention coefficient), visco-elastic moduli, loss tangent, and bread firmness increased proportionally with the amount of AF used.

The optimization of twenty-four responses allowed us to obtain, for each particle size, the optimal amaranth flour level which can be substituted into wheat flour to obtain dough and bread with the best rheological and technological properties. It can be concluded that particle size is very important for obtaining the desired rheological properties together with the appropriate bread volume and firmness. These optimal composite flours do not alter the dough network but improve the technological parameters desired to enrich bread and other bakery products due to particle size and approximate composition.

**Author Contributions:** Conceptualization, I.C. and S.M.; methodology, I.C.; software, I.C.; validation, I.C. and S.M.; formal analysis, I.C. and S.M.; investigation, I.C.; resources, I.C. and S.M.; data curation, I.C. and S.M.; writing—original draft preparation, I.C.; writing—review and editing, I.C. and S.M.; visualization, I.C. and S.M.; supervision, S.M.; funding acquisition, S.M. The authors contributed equally to this research. All authors have read and agreed to the published version of the manuscript.

**Funding:** This research was funded by "Stefan cel Mare" University of Suceava.

**Institutional Review Board Statement:** Not applicable.

**Informed Consent Statement:** Not applicable.

**Data Availability Statement:** Not applicable.

**Acknowledgments:** This work was funded by the Ministry of Research, Innovation and Digitalization within Program 1—Development of national research and development system, Subprogram 1.2—Institutional Performance—RDI excellence funding projects, under contract no. 10PFE/2021.

**Conflicts of Interest:** The authors declare no conflict of interest.

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
