# Peer review of "Features of Bread Made from Different Amaranth Flour Fractions Partially Substituting Wheat Flour"

_applsci, doi:10.3390/app12020897_

Round 1
Reviewer 1 Report
Dear Authors, the article is interesting and well written. Below are my remarks and comments.
Line 22: gelatinization, and retrogradation,
Line 51-53: this is well-known information, however, please provide references
Line 58: flour dilutes the gluten content in wheat flour which adversely affects the rheological
Line 75: This study aimed to optimize the formulation of amaranth flour addition level
Line 81: whether amaranth-wheat flour blend may be contemplated instead of amaranth-wheat composite flour
Line 84: I propose: Wheat flour (flour yield 65%) from S.C. MOPAN S.R.L., SUCEAVA, 84 ROMÂNIA was used.
Line 87: moisture content, was
Line 88: ash content, was determined
Line 106: composite flour blend
Line 108: making sourdough leaven from
Line 111: The sourdough leaven
Line 115-116: ….process, the dough was cut in 400 g/piece, molded, placed in aluminum trays for one hour to produce the final fermentation,…… Was the final fermentation time the same (one hour) for each of the dough fomulations ? How was it checked whether the dough reached its optimal growth (end of final fermentation)?
Line 126: The Falling number index (FN) of wheat flours and flour blends……Whether the falling number was converted to the 14% moisture of the flour.
Table 2: I suggest adding the line Amaranth flour, above the line the Particle size
Is it possible to add a table with the results of the chemical composition of naphtha? The text discusses the results, but they are not presented in the table.
Similarly, the tables do not present the other results obtained with the methods described in subchapters 2.3-2.5.
Noteworthy is the very well-planned analysis of the results and its clear graphic presentation.
Line 417-418: Please also consider the fiber content of amaranth flour and its influence on the formation of the gluten dough structure.
Line 470-486: Can the impact of damage to starch granules in wheat flour and amaranth flour on the quality of the dough used be taken into account?
Line 511: Amaranth flour addition
Line 532: In my opinion, it would also be useful to present the estimated chemical composition of amaranth-wheat bread obtained from a flour blend of optimal composition and comparison with the estimated chemical composition of wheat (control) bread.
Author Response
Dear reviewer, thank you for reading the manuscript carefully, and for your corrective comments and suggestions, which helped us improve the manuscript. We now have changed our manuscript according to your suggestion.
Comment to the Author
Dear Authors, the article is interesting and well written. Below are my remarks and comments.
Line 22: gelatinization, and retrogradation,
We deleted the comma.
Line 51-53: this is well-known information, however, please provide references
We added a reference.
Line 58: flour dilutes the gluten content in wheat flour which adversely affects the rheological
We added the word “flour”.
Line 75: This study aimed to optimize the formulation of amaranth flour addition level
We added the word “flour”.
Line 81: whether amaranth-wheat flour blend may be contemplated instead of amaranth-wheat composite flour
We modified how you kindly suggest.
Line 84: I propose: Wheat flour (flour yield 65%) from S.C. MOPAN S.R.L., SUCEAVA, 84 ROMÂNIA was used.
We modified how you kindly proposed.
Line 87: moisture content, was
We deleted the comma.
Line 88: ash content, was determined
We deleted the comma.
Line 106: composite flour blend
We replaced the word “composite” with “blend”.
Line 108: making sourdough leaven from
We replaced the “sourdough” with “leaven”.
Line 111: The sourdough leaven
We replaced the “sourdough” with “leaven”.
Line 115-116: ….process, the dough was cut in 400 g/piece, molded, placed in aluminum trays for one hour to produce the final fermentation,…… Was the final fermentation time the same (one hour) for each of the dough fomulations ? How was it checked whether the dough reached its optimal growth (end of final fermentation)?
Conditions for the fermentation process were the same for each dough formulation. The optimal growth (end of final fermentation) was searched by previous attempts, noticing when they doubled their volume.
Line 126: The Falling number index (FN) of wheat flours and flour blends……Whether the falling number was converted to the 14% moisture of the flour.
We added the information, as you kindly suggest.
Table 2: I suggest adding the line Amaranth flour, above the line the Particle size
We added the line Amaranth flour, as you kindly suggested.
Is it possible to add a table with the results of the chemical composition of naphtha? The text discusses the results, but they are not presented in the table.
The chemical composition of amaranth fractions and wheat flour is presented in Table 2.
Similarly, the tables do not present the other results obtained with the methods described in subchapters 2.3-2.5.
Thank you for your observation. The results are visible in Tables 3 and 4, according to the regression models presented.
Noteworthy is the very well-planned analysis of the results and its clear graphic presentation.
Thank you very much for appreciating our work.
Line 417-418: Please also consider the fiber content of amaranth flour and its influence on the formation of the gluten dough structure.
Thank you for your suggestion. We completed with information regarding the influence of amaranth fiber content on the gluten dough structure.
Line 470-486: Can the impact of damage to starch granules in wheat flour and amaranth flour on the quality of the dough used be taken into account?
Thank you for your deeply reading and careful observation. We completed with more information and new references.
Line 511: Amaranth flour addition
We completed.
Line 532: In my opinion, it would also be useful to present the estimated chemical composition of amaranth-wheat bread obtained from a flour blend of optimal composition and comparison with the estimated chemical composition of wheat (control) bread.
Thank you very much for your kind suggestion. The subject is approached in another work which includes a complex characterization of the bread nutritional profile obtained after the optimization process in comparison with the control bread.
Reviewer 2 Report
This research studied the influence of amaranth flour particle size and addition level on the wheat dough rheological properties and on technological and textural bread properties. The authors determined the optimal flour-amaranth composition according to the technological properties. The only thing missing is a sensorial characterization to evaluate the characteristics of the final product also regarding its taste. Anyway considering also the field of research of this journal the work is very valuable. The experimental part is accurately done and results are well-presented bringing to solid conclusions. In the attached file authors can find some comments I did on the manuscript. In particular, some parameters and values chosen to conduct some measurements need to be justified. I also suggest substituting some references which are not very recent with recent literature, especially regarding rheological characterization. Additionally, please check the journal template because the mandatory section "Featured Application" is missing for example. Other minor comments are specified in the attached document.

Author Response
Dear reviewer, thank you for reviewing our work. We appreciate very much the careful review and comments being able to clarify some parts of the manuscript and improve it. We tried to revise the comments and we respond to all your comments.
Comment to the Author
This research studied the influence of amaranth flour particle size and addition level on the wheat dough rheological properties and on technological and textural bread properties. The authors determined the optimal flour-amaranth composition according to the technological properties. The only thing missing is a sensorial characterization to evaluate the characteristics of the final product also regarding its taste. Anyway considering also the field of research of this journal the work is very valuable. The experimental part is accurately done and results are well-presented bringing to solid conclusions. In the attached file authors can find some comments I did on the manuscript. In particular, some parameters and values chosen to conduct some measurements need to be justified. I also suggest substituting some references which are not very recent with recent literature, especially regarding rheological characterization. Additionally, please check the journal template because the mandatory section "Featured Application" is missing for example. Other minor comments are specified in the attached document.
- Featured Application is missing.
We completed the mandatory section “Featured Application”, as you kindly suggested.
- Key words: I suggest to us amaranth flour instead amaranth.
We completed.
- Line 39-43: the sentences are hard to read. Please rephrase.
We revised those sentences as you kindly suggest.
- References are not very recent. I suggest to substitute with more recent references.
We deleted the old reference, and we substituted it with recent references.
- Line 102: Delete comma
We deleted the comma.
- Lines 107-108: Why did you choose this recipe?
The recipe was searched by previous attempts, in order to improve the non-gluten flour properties, for better-leavened dough.
- Line 156: How did you choose this values?
We make a mistake and we corrected the values.
- The only thing missing is a sensorial characterization to evaluate the characteristics of the final product also regarding its taste.
Thank you very much for your kind suggestion. The subject is approached in another work which includes a complex characterization of the nutritional and sensorial profile of the bread obtained after the optimization process in comparison with the control bread.
- In particular, some parameters and values chosen to conduct some measurements need to be justified.
We completed and modified the parameters and values chosen.